# Human Placental Schistosomiasis—A Systematic Review of the Literature

**DOI:** 10.3390/pathogens13060470

**Published:** 2024-06-03

**Authors:** Jacob Gerstenberg, Sasmita Mishra, Martha Holtfreter, Joachim Richter, Saskia Dede Davi, Dearie Glory Okwu, Michael Ramharter, Johannes Mischlinger, Benjamin T. Schleenvoigt

**Affiliations:** 1Institute for Tropical Medicine, Eberhard-Karls University Tübingen, 72074 Tübingen, Germany; jacob.gerstenberg@posteo.de; 2Centre de Recherches Médicales de Lambaréné, Lambaréné 242, Gabon; 3Department of Obstetrics and Gynecology, Heidekreis Klinikum, 29664 Walsrode, Germany; 4Tropical Medicine Unit, Department of Gastroenterology, Hepatology and Infectious Diseases, Faculty of Medicine, Heinrich-Heine-University, Moorenstr. 5, 40225 Duesseldorf, Germany; 5Institute of International Health, Global Health Center, Charité University Medicine, 13353 Berlin, Germany; 6Swiss Tropical and Public Health Institute, 4123 Allschwil, Switzerland; 7Center for Tropical Medicine, Bernhard Nocht Institute for Tropical Medicine & I. Department of Medicine, University Medical Center Hamburg-Eppendorf, 20359 Hamburg, Germany; 8German Centre for Infection Research (DZIF), Partner Site Hamburg-Luebeck-Borstel, 20359 Hamburg, Germany; 9Institute of Infectious Diseases and Infection Control, Jena University Hospital, Friedrich-Schiller-University, 07747 Jena, Germany

**Keywords:** schistosomiasis, placenta, adverse birth outcomes, pregnancy, praziquantel

## Abstract

Background: Schistosome egg deposition in pregnant women may affect the placenta of infected mothers and cause placental schistosomiasis (PS). Histopathological examination of placental tissue is an inadequate detection method due to low sensitivity. So far, there has not been any systematic review on PS. Methods: We conducted a systematic literature search on PubMed, EMBASE, and Medline and included all publications that reported microscopically confirmed cases of PS, as well as the relevant secondary literature found in the citations of the primarily included publications. Results: Out of 113 abstracts screened we found a total of 8 publications describing PS with a total of 92 cases describing egg deposition of dead and/or viable eggs and worms of *S. haematobium* and *S. mansoni* in placental tissue. One cross-sectional study investigating the prevalence of PS and its association with adverse birth outcomes, found 22% of placentas to be infested using a maceration technique but only <1% using histologic examination. Additionally, no direct link to deleterious pregnancy outcomes could be shown. Conclusions: PS is a highly unattended and underdiagnosed condition in endemic populations, due to a lack of awareness as well as low sensitivity of histopathological examinations. However, PS may play an important role in mediating or reinforcing adverse birth outcomes (ABO) such as fetal growth restriction (FGR) in maternal schistosomiasis, possibly by placental inflammation.

## 1. Introduction

Schistosomiasis is a parasitic disease affecting around 220 million people worldwide [1], among which there is an estimated 40 million women of reproductive age [2]. After infecting humans, *Schistosoma* spp. excrete eggs that can be trapped in various tissues, depending on the causative species. In women, if trapped within the genital organs, they can cause female genital schistosomiasis (FGS) [3]. Pregnant women may also be affected by egg deposition in the placenta. Previous research describes the placental villi, the intervillous space, and the decidua, to be the most common places of detection [4]. Pregnant women are a particularly vulnerable population for schistosomiasis, as it contributes to maternal anemia [5] and may be linked to adverse birth outcomes (ABO) in their newborns, such as low birth weight, small for gestational age (SGA), fetal growth restriction (FGR) and stillbirth [6,7]. The exact mechanisms of the pathophysiological link between maternal schistosomiasis and ABO have not yet been fully understood and might be multi-causal. Suggested contributors include maternal anemia, a known risk factor for ABO [8], which is associated with schistosomiasis [5] as well as other parasitic diseases. Furthermore, inflammation at the feto-maternal interface due to placental egg deposition and the trans-placental transfer of schistosomal antigens may play an important role [6]. Since the beginning of the 20th century, there have only been a few case reports and clinical trials investigating placental schistosomiasis (PS). This may be due to neglected attendance to the topic, but also methodologic difficulties to identify infested placentas with sufficient sensitivity, because standard histopathological methods can only investigate a little part of the placenta. Sutherland et al. already called in 1965 for studies using tissue dissolution techniques, and investigated a series of placentas with a maceration technique using 10% sodium hydroxide [9]. Renaud et al. then later used in 1972 a maceration technique using potassium hydroxide (KOH) [10]. Recently, a description for an improved maceration technique has been described by Holtfreter et al. (2017), using ethanol fixation and maceration with 4% KOH over 24 h, which allowed a complete dissolution of placental tissue while maintaining well-preserved morphological features of the observed schistosomal eggs [11]. Because of the high underestimation and underdiagnosing of PS, we aimed to review the current literature and give an overview of the microscopically confirmed cases that have been described in the past century. We also aimed to discuss the role of PS as a possible mechanistic link between maternal schistosomiasis and ABO.

## 2. Materials and Methods

A systematic literature search was conducted in PubMed, EMBASE, and MEDLINE. The search terms used to gather information on PS were (placenta OR placental) AND (schistosomiasis OR schistosoma OR bilharzia). A total of 113 listed publications were screened for information on PS published in the years 1957 to 2024. Furthermore, we included literature that was published between 1948 and 1994 that was not accessible within the above-mentioned electronic databases but cited in the reviewed literature. Only publications including microscopically confirmed cases of PS in humans were eligible and are displayed in the table and flow chart, respectively. Eligibility was assessed by two reviewers independently and potential discrepancies were resolved by a third reviewer. The risk of bias was assessed by a tool using eight items evaluating the quality of selection, ascertainment, causality, and reporting of each publication, which has been proposed by Murad et al. (2018) [12]. Based on this tool an overall statement of the methodological quality was made by two reviewers for each eligible publication, respectively. Potential discrepancies were again resolved by a third investigator. Details of the risk of bias assessment can be found in the Appendix A. This systematic review follows the recommendations of “Preferred Reporting Items for Systematic reviews and Meta-Analyses” (PRISMA). The PRISMA checklist and the review protocol can be found in the Appendix A.

## 3. Results

Of the 113 publications screened, 3 were trials without microscopic confirmation of PS, 3 were case reports on schistosomiasis in pregnancy without confirmation of PS, 11 were trials on schistosomiasis without aspects of PS, 34 addressed immune responses and inflammation associated with schistosomiasis in humans, 20 were animal models, 6 were in vitro studies, 14 were reviews, and 18 were publications without any aspects of schistosomiasis. Those 109 publications were excluded. The remaining four publications identified in PubMed, EMBASE, and MEDLINE were case reports that referred directly to placental involvement in schistosomiasis. In addition, four publications reporting on PS were identified in the references of the listed literature. For completeness, it is worth mentioning that an additional conference abstract from 1968 was identified that addressed PS but did not provide clear information about additional cases. Overall, eight scientific full-text publications on the placental involvement of schistosomiasis were identified and considered for this review (Figure 1).

In eight publications that were considered relevant for this review, we found 92 documented cases of PS, which are summarized in Table 1. Among these 92 cases, 4 cases reported adult worms and all cases schistosome eggs in placental tissue, respectively. In 84 cases, the causative species was *S. haematobium*, in 7 cases *S. mansoni,* and 1 case showed placental infestation of both species. None of the cases described a direct detection of *S. japonicum* eggs or worms in the placenta.

The first mention of intrauterine infection with *Schistosoma* spp. dates to Fujinami and Namakura in 1911, who detected *S. japonicum* in the fetus of an infected pregnant dog [19]. The experimental intrauterine infestation of animals with *Schistosoma* spp. was later confirmed by Narabayashi et al. in 1914 [20]. Narabayashi also first described egg deposition in the placenta of two guinea pigs. He then described a case series of 22 human newborns, whose mothers were exposed to *S. japonicum* during pregnancy, of which three infants were infected with *S. japonicum* despite never having had contact with fresh water [20]. Some concluded that this could be seen as proof that vertical transmission of the parasite could also happen in humans [21]. However, transplacental invasion of *Schistosoma* spp. has never been confirmed in humans since then. About 30 years later, Prates (1948) first mentioned the placental involvement of schistosomiasis in humans in a case series of five women due to an infection with *S. haematobium* [13]. 

The first detailed histopathological description of human PS was later provided by Sutherland et al. in 1965, who described a case with placental egg infestation of *S. haematobium* in a pregnant woman with epileptiform seizures [9]. The authors realized the need for a maceration technique as a method of choice for the investigation of placental tissue because random sectioning of the placental tissue would give false negative results. After their observation of the initial case, the authors further investigated another 22 placentas (using a 10% sodium hydroxide maceration) of which they found an additional 7 to be infected with either *S. haematobium* or *S. mansoni*. In 1968, Rodrigues presented several cases of PS at the Congress of the International Academy of Pathology in Milano, Italy [22]. However, the conference abstract leaves it unclear whether the cases mentioned were newly described cases from Mozambique or references to the existence of such cases from the scientific literature.

In 1969, Bittencourt et al. published one case of PS due to *S. mansoni* describing both egg deposition as well as adult worms in the examined placental tissue [14]. In 1980 the same author published another case series of four cases, including the previous case from 1969, and reported an additional three cases of stillbirths with granulomatous placentitis that might have been caused by placental infestation with *S. mansoni* [16]. Two years earlier in 1978, Viggiano et al. had reported another case of PS due to *S. mansoni* [15], but neither Viggiano nor Bittencourt performed a maceration technique to examine placentas. 

The most extensive investigation on PS until the present was conducted by Renaud et al. in 1972, who were the first to carry out a systematic investigation of placentas on an extensive scale in the Ivory Coast using a maceration technique with caustic potash. From the examined 322 placentas, 72 (22.3%) were positive for schistosome eggs (*S. haematobium*) by maceration technique whereas histopathologic sectioning revealed only 1 positive result showing eggs in the intervillous space [10]. None of the women were previously known to be infected with *S. haematobium*. 

In 1994, Peres et al. published another case report in which he described a woman with a prolapsed umbilical cord who subsequently gave birth to an otherwise healthy premature baby via cesarean section. Routine examination of the placenta by histological examination revealed both dead and viable eggs of *S. mansoni* in the villous vessels without surrounding inflammatory reaction [17]. The most recent description is from 2014 by Schleenvoigt et al. who first described PS in a German traveler returning from Malawi. The woman had presented with painless macrohematuria and was diagnosed and treated for schistosomiasis during pregnancy. The histopathological examination of the placenta 18 weeks post-treatment revealed egg deposition of *S. haematobium* inside the villi.

## 4. Discussion

### 4.1. Underestimated Prevalence of Placental Schistosomiasis?

Placental schistosomiasis has recently been reaffirmed to be underestimated and underdiagnosed, mainly due to the lack of attention to this issue [11]. This seems to be confirmed by the latest findings from Franz et al., who examined 268 placentas of women from the Ivory Coast and Ghana with PCR for the presence of *Schistosoma* spp. and found positive signals in 19% (*n* = 51) of the samples [23]. The latter is in accordance with the earlier reported prevalence of PS of 20% in an endemic population of the Ivory Coast by Renaud et al. [10] detected by maceration. However, the evidence available to date on the true prevalence of PS is scarce.

### 4.2. Contribution to Adverse Birth Outcomes (ABO)

The growing importance of PS is not only due to its unacknowledged occurrence but also due to its possible role as a mechanistic link between maternal schistosomiasis and adverse birth outcomes (ABO), such as prematurity, low birth weight (LBW), fetal growth restriction (FGR), and stillbirth or miscarriage [6]. Of the 92 cases summarized in Table 1, 22 were associated with prematurity, 22 with LBW, and 13 with fetal death (miscarriage or stillbirth). However, there was most likely a strong selection bias, as the examination of the placenta in most case reports was due to pregnancy complications. Only the cross-sectional study of Renaud et al. in 1972 investigated independently a possible association between PS and ABO. The analysis of newborn weight and length revealed no association with PS. Furthermore, these authors observed that placental infestation generally appeared to take place after the 3rd month of gestation, probably due to circulatory changes in the pelvic blood system throughout the pregnancy, and concluded that PS seems not to contribute to fetal death, at least before this time point in pregnancy [10]. Nevertheless, there remains evidence, albeit contradictory, that maternal schistosomiasis contributes to ABO, particularly the occurrence of LBW [24,25,26,27]. The underlying pathophysiologic mechanisms are not yet fully understood. Maternal anemia, which is strongly associated with schistosomiasis [5] is a known risk factor for ABO [8], but cannot explain sufficiently all contributions [7]. Placental egg deposition and trans-placental transfer of schistosomal antigens to the fetal bloodstream, both closely related to PS, may play an important role as well [6]. 

### 4.3. Discussed Pathophysiologic Mechanisms

The transfer of schistosome soluble egg antigen (SEA) across the placenta has been well documented [28,29,30] and appears to happen frequently (up to 86%) in maternal schistosomiasis [30]. McDonald et al. investigated the effects of SEA on placental trophoblast cells in vitro and found a dose-depending upregulation of proinflammatory and chemotactic cytokines [31]. Kurtis et al. have shown that maternal schistosomiasis due to *S. japonica* was directly associated with both elevated proinflammatory signals in the placenta (TNF-α, IL-6, IL, IL-1β, and TNF-IIR) and fetal cord blood (TNF-IIR and IFN-y), respectively [32]. Considering that the upregulation of fetal proinflammatory signals seems to be dose-depending, one can assume that direct deposition of eggs in the placenta might elicit a much stronger proinflammatory response. Increased levels of endotoxins circulating in the maternal bloodstream due to schistosomiasis-derived microbial translocation may also contribute to placental inflammation [33]. Elevated levels of endotoxins have been found in both non-pregnant [34] and pregnant adults and were associated with placental inflammation [33]. 

The immune milieu at the fetal–maternal interface plays a crucial role in various aspects of healthy pregnancies and is physiologically delineated by a cytokine profile linked to TH-2 cells [35,36]. However, its disbalance through maternal infections can lead to pregnancy complications [37]. For example, shifts towards a more proinflammatory cytokine imprint of the placenta due to maternal malaria infection have been linked to intrauterine growth restriction (IUGR) [38]. Also, in the aforementioned study by Kurtis et al., elevated levels of placental TNF-α and IL-1β were associated with decreased birth weight [32]. Similar findings were reported by Abioye et al., indicating that increased levels of proinflammatory cytokines were linked to a higher risk of prematurity and small for gestational age (SGA) infants [39].

### 4.4. Treatment Strategies

The question remains whether maternal schistosomiasis or PS specifically should be treated during pregnancy. Praziquantel, the drug of choice for schistosomiasis, has been withheld from pregnant women for decades, due to safety concerns. This practice changed with an updated risk–benefit analysis of the WHO in 2002 [40]. The most recent WHO treatment guidelines for schistosomiasis in 2022 recommend the treatment of pregnant women in the second and third trimesters both on an individual and mass drug administration level [41]. Still, the use of praziquantel in pregnancy remains a neglected field of research [2]. Two major randomized controlled studies have addressed the safety and efficacy of praziquantel in pregnant women infected with *S. mansoni* [42] and *S. japonicum* [43], respectively. Both trials showed that praziquantel can be safely used during the second or third trimester. A more recent trial carried out in Gabon addressed the same question in pregnant women infected with *S. haematobium* (clinicaltrials.gov NCT03779347) [44]. So far, no trials have investigated the treatment effects of praziquantel stratified among women who were suffering from PS. 

## 5. Conclusions

We have reviewed and summarized the available literature concerning PS and its contribution to ABO as well as possible underlying mechanisms. The determination of the frequency of PS among pregnant women in endemic regions remains a challenge, and further research using maceration techniques [11] is required. Further investigation on the association of PS on ABO and possible advantages of treatment are also urgently needed to prevent morbidity in both mothers and their offspring. Future research on ABO should not be restricted to crude birthweight and prevalence of LBW, but include the assessment of being small for gestational age and calculate the latter with sonographic confirmed gestational age and standard international growth curves [45].

## Figures and Tables

**Figure 1 pathogens-13-00470-f001:**
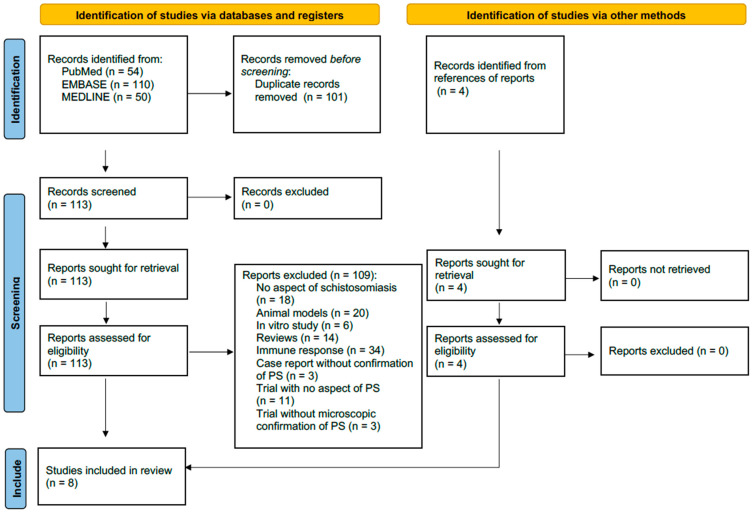
PRISMA flowchart of the systematic literature search. The search terms used for identification of records were (placenta OR placental) AND (schistosomiasis OR schistosoma OR bilharzia).

**Table 1 pathogens-13-00470-t001:** Overview of the cases of placental schistosomiasis found in the selected literature.

Year	Author	Genre	No. of Cases	Country of Origin	Age	*Schistosoma* spp.	Adult Helminth	Pregnancy Complications	Applied Technique	Risk of Bias
1948	Prates [13]	Case series	5	Mozambique	-	*S. haematobium*	no	NA	Histology	Poor
1965	Sutherland [9]	Case series	8	South Africa	19, NA	*S. haematobium* (6) *S. mansoni* (1) both species (1)	no	Seizures (1) Preeclampsia (1)	Histology (1), Maceration (7)	Good
1969	Bittencourt [14]	Case report	1	Brazil	34	*S. mansoni*	yes	No	Histology	Good
1972	Renaud [10]	Clinical study	72	Ivory Coast	-	*S. haematobium*	no	LBW (21) Fetal death (10) Prematurity (21)	Histology and Maceration	Good
1978	Viggiano [15]	Case report	1	Brazil	19	*S. mansoni*	yes	Fetal death	Histology	Good
1980	Bittencourt [16]	Case series	4 (3) ^Ψ^	Brazil	34, 23, 18, NA	*S. mansoni*	yes (2)	Fetal death (2) EUG (1)	Histology	Good
1994	Peres [17]	Case report	1	Brazil	31	*S. mansoni*	no	Prematurity, LBW	Histology	Good
2014	Schleenvoigt [18]	Case report	1	Malawi	28	*S. haematobium*	no	no	Histology	Good

^Ψ^ One of the four cases was the previously described case report from 1969 by the same author. EUG = extrauterine gravidity.

## Data Availability

No new data were created in this article.

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
