# Peer review of "Human Placental Schistosomiasis—A Systematic Review of the Literature"

_pathogens, 2024, doi:10.3390/pathogens13060470_

Round 1

Reviewer 1 Report

Comments and Suggestions for Authors

The manuscript submitted by the author entitled “Human Placental Schistosomiasis – A Review of the Literature” is a review which aimed to give an overview of the past cases concerning Placental Schistosomiasis and pregnancy in a current literature review. This study is important in its field, but the paper is not well-written/structured, and the sampling efforts is not enough. I would recommend to the authors to deeply revise the manuscript to address the critical issues listed below.

General comments

1.    Overall, the manuscript is well written. 

2.    But there are some typos and minors test editing

3.   The study is of great interest but the objective of the study is not clearly presented by the authors and not aligned with the conclusions

4.   Authors have titled their manuscript “A Review of the Literature”, which is not but mostly presented as a Systematic Review. And for that reason; more details are needed in the methods section. 

5.  This is unclear how the selection strategy was conducted, how the data extraction was organized (for example by how many investigators); for how data were curated (for example duplicates especially when different databases are used); how bias where manage; how the quality of evidence was judged.

6.    To be more precise, authors should also get in contact with authors of those publications to know if there are some results not presented in their publications.

7.    The paper needs a better structure.

Introduction

Line 43: replace “become” by “can be”. Most of the eggs are excreted in urine and stool.

Line 65: Replace “our study group” by the authors’ names and year.

Line 68-71: “Because of … century.”. This objective is unclear to me. The rational of conducting this study is not clear.

Materials and Methods

Line 73: “PubMed was…” Other database should be used (EMBASE, Web of Sciences and , Google Scholar).

Line 74: Why using only ‘schistosomiasis’ to search. It might be worth to try other spelling or parasite’s names as ‘Bilharzia’ or ‘Schistosoma’.

Line 77-79: “Only publications … respectively.”. In my opinion the objective of the study should be organize around this.

Line 79-91: “Of the 47 … (figure 1).”. This is the result of the screening, not the methodology.

Line 84: authors stated, “Those 45 publications were excluded”. But only 44 are counted from line 93: This flowchart is a result. 

Results

The result section looks like a discussion. The authors included in their selection criteria the ‘anatomopathological component’, but this is not clearly shown in the results.

Line 104-105: This is not in the table, is this a result or the discussion?

Line 105: Replace “a” by “an” in “a infected pregnant dog”.

Table 1: replace “1971” by “1972”at the fourth line resuming the results of Renaud et al.

Line 131: Replace “1965” by “1969”.

Discussion

There are mixture of result and discussion in the 4.2 and 4.3 sections.

Conclusion

Line 226: “further research using improved maceration techniques”. Why? Not clear from the results.

Reviewer 2 Report

Comments and Suggestions for Authors

Schistosomiasis, also known as snail fever, is a disease caused by flatworms of the genus Schistosoma. The urinary tract and/or intestines may become infected.

The disease is transmitted through contact with freshwater contaminated with parasites, which are released from infected freshwater snails. The disease is especially common among children in developing countries as they are more likely to play in contaminated water. Other high-risk groups include: farmers, fishermen or people who use dirty water for daily use. It belongs to the group of helminth infections and the diagnosis is through the identification of parasite eggs in the person's urine or feces. It can also be confirmed by identifying antibodies against the disease in the blood.

Disease prevention methods include improving access to clean water and reducing snail numbers. In areas where the disease is common, the medicine praziquantel can be given once a year, for the entire group. This is done to reduce the number of infected people and, consequently, the spread of the disease. Praziquantel is also the treatment recommended by the World Health Organization (WHO) for those who are infected.

During pregnancy, schistosomiasis can occur and can affect the placenta of infected women and cause placental schistosis (PS). The histopathological examination of the placental tissue is an inadequate detection method due to low sensitivity. This manuscript submitted by Jacob Gerstenberg and collaborators was worked on secondary literature and the authors also found a total of 8 publications describing SP with a total of 92 cases, describing the deposition of dead eggs and/or viáveis and worms of S. haematobium and S. mansoni in the placenta of infected women.

A cross-sectional study investigating the prevalence of PS and its association with adverse birth outcomes found that 22% of placentas were infested using a maceration technique, but only <1% using histological examination. Além disso, no direct ligation with deleterious results of pregnancy can be demonstrated. This manuscript concludes that PS is a highly neglected and underdiagnosed condition in endemic populations, due to lack of knowledge and low sensitivity of two histopathological examinations. However, PS can play an important role in mediating or reinforcing adverse birth outcomes (ABO), such as fetal growth restriction (FGR) in maternal schistosis, possibly due to placental inflammation.

Extremely interesting review for the study area.

Reviewer 3 Report

Comments and Suggestions for Authors

Dear Authors,

This is a very nice Review article summarizing human placental Schistosomiasis that is one of the Neglected Tropical Diseases (NTDs).
As mentioned in your manuscript, PS has not been carefully investigated even it may have an important role in adverse birth outcomes (ABO).

I only have very minor concerns to your manuscript.
1. (line 83) "eight were animal models" would be "nine were animal models".
2. (line 175) "adverse birth outcomes" would be "ABO".
3. Please check "References" carefully. All species names should be in italics.

Thank you.

Round 2

Reviewer 1 Report

Comments and Suggestions for Authors

The authors' responses and changes fully address the points raised when revising the manuscript.

I just wonder that using other database or trying other spelling or parasite's names did not increase the number of relevant documented cases of PS for this systematic review. I would like the authors to confirm that aspect.